# It's dark under the lamp? The moderating role of executives' accounting competence on relationship between goodwill impairment signal and goodwill impairment

Qiandan Deng[ID]*

Business School of Sichuan Normal University, Chengdu, Sichuan Province, China

* dengqiandan@126.com

**Data Availability Statement:** All relevant data are within the paper.

**Funding:** This article is funded by Sichuan Normal University Research Fund (Project No.: 22XW045).

## Abstract

This study examines the influence of executives' accounting competence on relationship between goodwill impairment signal and goodwill impairment, considering the perspective of performance compensation commitment. The research employs an empirical research method and utilizes a sample of A-share listed companies in China that have signed performance compensation commitment agreements from 2007 to 2022. I found that the executives' accounting competence weakens the relationship between goodwill impairment signal and goodwill impairment. The stronger the executives' accounting competence is, the weaker the goodwill impairment signal is, that is, the probability of the goodwill impairment is lower and the scale of goodwill impairment is smaller. Further research shows that the negative effect of executives' accounting competence is more significant when the performance compensation commitment of the target assets is between 80% and 100%. Through heterogeneity analysis, it is found that the negative effect of executives' accounting competence will be suppressed when the sample is in the period of equity incentive or the auditor audits the key issues of goodwill impairment. The research in this paper not only enriches the relevant literature on the characteristics of executives, but also discusses the subsequent measurement of goodwill, which is also conducive to promoting the formulation of accounting standards and assisting the supervision of capital markets, and has important theoretical and practical significance.

## 1 Introduction

Goodwill is generated during the process of enterprise merger and reorganization, and the accounting treatment method for goodwill has shifted from systematic amortization to impairment testing. Standard-setters argue that impairment testing can offer more tailored information regarding goodwill compared to yearly amortization, thereby enhancing the value of financial statement information [1–3]. The impairment test based on fair value estimation entails a multitude of parameters, incorporating numerous hypothetical conditions. Executives

The funders had no role in study design, data collection and analysis, decision to publish, or preparation of the manuscript.

**Competing interests:** The authors have declared that no competing interests exist.

are afforded increased discretion in establishing provisions for the impairment of goodwill. This discretion renders the impairment information associated with goodwill unverifiable, thereby diminishing the quality of financial accounting information [4]. The substantial amount of goodwill impairment among numerous listed companies in recent years has significantly impacted performance, drawing the attention of all stakeholders. In order to explore ways to restrain managers' opportunistic behavior in the provision of goodwill impairment, I found the signal to measure the risk of goodwill impairment. According to my past research, the realization degree of performance compensation commitment can be used as a signal of whether goodwill needs to be depreciated. Because of the performance compensation commitment is an agreement signed by both parties to the M&A for the performance of the M&A target in the next 3–5 years, and goodwill is generated based on the excess profitability of the merged assets. If the future performance of the target assets declines and the excess profitability declines, the goodwill should be depreciated. Therefore, the realization degree of performance compensation commitment can be used as a signal of whether goodwill needs to be depreciated, that is, the higher the unfulfilled performance compensation commitment of the underlying assets, the higher the probability of impairment of goodwill, and the larger the scale of impairment of goodwill. Albrecht, Mauldin and Newton have found that when executives are motivated to disclose false information, executives with strong accounting ability and rich financial reporting experience will increase the possibility of material misstatement. At this point, the dark side of executives' accounting competence will appear in the presence of some incentive factors [5]. This situation may exacerbate when executives' competence (especially accounting competence) is stronger. Goodwill impairment can diminish the net profit of listed companies, significantly impacting their current performance. Given that executives' interests are typically aligned with the company's performance, they may be motivated to intervene in goodwill impairment to further their own benefit [6–8]. Hence, the executives have strong accounting competence, they may use more subtle methods to manipulate impairment tests and affect the signal function of performance compensation commitment. The opportunistic behavior of executives will lead to the decrease of the information quality of financial reports [9]. Therefore, it is very important to discuss the role of excellent accounting competence of executives on relationship between goodwill impairment signal and goodwill impairment and how to alleviate this problem. By examining the impact of executives' accounting competence on relationship between goodwill impairment signal and goodwill impairment, I can assess whether executives' strong accounting competence raises the likelihood of opportunistic behavior in the context of goodwill impairment. When the corporate governance mechanism is more effective, executives are less likely to act opportunistically and their discretion is more limited [10–12]. Therefore, considering the corporate governance mechanism, I examine strategies to mitigate executives' opportunistic behavior in recognizing goodwill impairment.

This study may have the following contributions. First, in a theoretical significance, it provides a new perspective to test the opportunistic behavior of executives to accrue goodwill impairment. The existing literature of effect of goodwill impairment is mainly carried out from the perspectives of macroeconomic factors and discretion, and has achieved rich research results. But less attention is paid to the influence of executives' competence (especially accounting competence) on goodwill impairment. This study enriches the research literature on the influencing factors of goodwill impairment. Secondly, it enriches the research literature on the characteristics of executives. Taking the performance compensation commitment as the breakthrough point, this paper examines the influence of executives' accounting competence on relationship between goodwill impairment signal and goodwill impairment and whether executives' accounting competence plays a positive or negative role, which expands

the boundary of the literature on the characteristics of executives' competence. Third, in a practical sense, it will help the regulatory agencies to strengthen the supervision of executives' behavior. Based on the impairment of goodwill in M&A, this paper points out that executives have the motivation, opportunity and competence to intervene in the impairment of goodwill for their own interests, and the accounting competence of executives may have a negative effect, suggesting that regulators need to strengthen supervision over executives' behavior, which is crucial for upholding the normal functioning of the capital market.

## 2 Literature review

### 2.1 Economic consequences of executives' competence

The existing literature research shows that the competence of executives plays an important role in enterprise decision-making [13, 14]. However, due to their different knowledge, expertise, skills and experience, the competence of executives varies greatly [13, 15]. Many scholars have studied this issue and obtained rich research results. Goldfarb and Xiao found that executives who possess greater experience and higher levels of education demonstrate a preference for entering markets characterized by a lower number of competitors [16]. Kor discovered that executives' accumulated experience in the industry helps the team to seize new growth opportunities [15]. CEO with industry expertise have excellent negotiation skills, usually negotiate better deals in M&A and pay a lower premium for the target [17]. Chen thought the competence of executives is an important part of the success of enterprise innovation, and executives with better skills can effectively transform innovative ideas into valuable new products [18]. The high competence of executives is also manifested in the assurance of earnings quality [19, 20], can increase the value of the company [14], can get a higher M&A premium, the company's share price performed better [18, 21], higher credit rating [22] and help companies avoid taxes [23]. Demerjian, Lev, Lewis and McVay found that the earnings quality of enterprises is positively related to executives' competence [19]. Yung and Chen found that high-competence executives are related to the increase of company value [14]. In M&A, when the CEO of the acquirer is more experienced in the target industry, the abnormal announcement return rate of the enterprise will be 1.2 to 2.0 percentage points higher [17]. In addition, some studies have studied the competence of executives from the perspective of manager change, and found that the competence of executives also has an impact on the capital market and the credit rating of enterprises. Demerjian, Lev and McVay found a positive correlation between the replacement of CEOs with low ability and the stock price response, while CEOs with higher ability were observed to enhance the company's performance after assuming office [21]. The capital market demonstrates a more positive valuation towards patents created by executives with higher capabilities [18]. Bonsall, Holzman and Miller found that with the improvement of executives' competence, the credit rating of enterprises will also be significantly improved, indicating that rating agencies pay attention to and consider the level of executives' competence when rating enterprises [22].

### 2.2 Influencing factors of discretion in goodwill impairment

Given the complexity of impairment testing, executives have considerable discretion in recognizing impairments, a view supported by extensive research. Beatty and Weber's study found that both before and after the introduction of SFAS142, executives exhibited discretion in accelerating or delaying recognition of impairments [6]. Impairment testing compares the present value of future cash flows with the book value, with standard-setters believing it can convey more private information about the company's cash flows. Ramanna and Watts study did not find motives related to conveying private information in companies showing signs of

impairment but discovered that agency motives influenced executives in recognizing impairment [7]. Research from a psychological perspective found that the more confident executives are, the less timely their impairment recognition [24]. Sapkauskiene, Leitoniene and Vainiusiene found that during economic crises, executives tend to use discretion, inclining toward excessive recognition of impairments to reserve future profits for the company [25]. Li and Sloan also found that due to managerial discretion, the timeliness of impairment recognition cannot be guaranteed, leading to the delay of impairment recognition [26]. As gatekeepers of the capital market, auditors' independence and professionalism make their audit reports highly reliable. Therefore, when engaging in opportunistic behavior, executives seek the support of auditors. Research has found a negative correlation between auditors' dismissal and decisions supporting impairment recognition [27, 28].

The existing literature presents a comprehensive analysis of the factors influencing the discretion in impairment recognition and has yielded substantial research outcomes. However, there is a scarcity of studies on the important factor of executives' competence, particularly in the context of accounting competence. The significance of executives in the daily management and operation of a business is self-evident. However, the darker side of high managerial ability can manifest under certain incentive factors [5]. This study investigates the role of executives' accounting competence in recognizing impairments, not only enriching the relevant literature on managerial characteristics and discussing issues related to subsequent impairment measurement, but also facilitating the formulation of accounting standards and aiding in capital market regulation. Therefore, this study holds substantial theoretical and practical significance.

## 3 Theoretical analysis and research hypotheses

### 3.1 Goodwill impairment signal, executives' accounting competence and goodwill impairment

In recent years, the increased occurrence of significant impairment of goodwill leading to corporate financial distress has brought about a heightened focus on goodwill impairment across various sectors. Regulatory authorities have specifically issued guidance, such as Document No. 8, targeting the risks associated with goodwill impairment. Additionally, auditors have imposed more stringent audit procedures on the recognition of goodwill impairment in listed companies, including categorizing goodwill impairment as a key audit matter. Faced with such intense regulatory scrutiny, executives, to avoid the negative consequences of opportunistic behavior being exposed, may engage in more covert opportunistic actions, making it difficult for external observation. Goodwill is generated based on the excess earnings potential of the acquired assets. If the future performance of the acquired assets deteriorates, leading to a decline in excess earnings potential, impairment of goodwill should be recognized. My previous research has indicated that the extent of performance-related compensation commitments serves as a signal for the need to recognize goodwill impairment. Specifically, a higher level of unfulfilled performance-related compensation commitments is associated with a greater probability and magnitude of recognizing goodwill impairment. Therefore, by observing the impact of executives' accounting competence on the relationship between goodwill impairment signal and goodwill impairment, I can ascertain whether superior accounting competence increases the likelihood of opportunistic behavior when recognizing goodwill impairment.

Generally, given the complexity of impairment testing, executives with stronger accounting competence should handle impairment issues more reliably, and their impact on impairment should be positive. Highly capable management possesses enhanced valuation abilities, which can mitigate the increased risk of goodwill impairment resulting from high premiums during

mergers and acquisitions. Furthermore, highly capable management also demonstrates stronger M&A integration capabilities, which can enhance the long-term operational performance of listed companies and consequently reduce goodwill impairment. Executives' accounting competence may have a positive impact on the relationship between goodwill impairment signal and goodwill impairment.

Based on the above analysis, this study proposes the following hypothesis:

H1a: The executives' accounting competence reinforces this relationship between goodwill impairment signal and goodwill impairment. When the executives' accounting competence is stronger, the unfulfilled performance compensation commitment is higher, the higher the probability and scale of recognizing goodwill impairment.

However, capability is multifaceted, and high capability may be a double-edged sword. Arif, Mustapha and Abdul found that CEOs' political power and CEOs with high structural and expert power have a significant detrimental effect on earnings quality [29]. When executives have the motive to engage in opportunistic behavior, the more capable executives are, the more covert their discretionary practices may become, making it more difficult for external observation and supervision. Due to the complexity of goodwill impairment, the focus should be more on the accounting competence of executives. Accounting competence itself does not bring negative impacts, it provides the capacity to produce reliable financial reports. Whether accounting ability has a positive or negative impact depends on the manager's own motives. Albrecht, Mauldin and Newton found when executives have the motive to disclose information incorrectly, executives with high accounting ability and extensive financial reporting experience may increase the likelihood of significant misreporting. In such cases, the adverse effects of executives' accounting competence may manifest under certain incentive factors [5]. However, since impairment recognition reduces a listed company's net profit, significant impairment can severely impact the company's current performance, and executives' interests are usually tied to the company's performance. Therefore, for their own benefit, executives may have the motive to engage in opportunistic behavior when recognizing impairments. In such cases, the more capable executives' accounting competence is, the greater their negative impact on the relationship between goodwill impairment signal and goodwill impairment.

Based on the above analysis, this study proposes the following hypothesis:

H1b: The executives' accounting competence weakens this relationship between goodwill impairment signal and goodwill impairment. When the executives' accounting competence is stronger, the unfulfilled performance compensation commitment is higher, the lower the probability and scale of recognizing goodwill impairment.

## 3.2 Executives' accounting competence, equity incentive and goodwill impairment

Previous research has revealed that management exhibits opportunistic behavior regarding equity incentives [30]. On one hand, management can enhance the likelihood of meeting performance conditions by preemptively lowering performance targets, such as issuing more pessimistic performance forecasts before the announcement of the equity incentive proposal or engaging in earnings management prior to the disclosure of the equity incentive plan. On the other hand, management can retrospectively meet performance conditions through earnings management, as evidenced by existing research that has identified a significant phenomenon of "performance condition manipulation" in companies implementing equity incentive plans.

Goodwill impairment affects the performance of listed companies and, consequently, the interests of management. The incentive conditions for company management typically lead them to lean towards avoiding goodwill impairment [27]. Therefore, I predict that compared to periods without equity incentives, executives with higher accounting ability are more likely to avoid impairing goodwill during equity incentive periods, indicating a stronger negative impact of executives' accounting competence on the relationship between goodwill impairment signal and goodwill impairment.

Based on the comprehensive analysis, this study further proposes the following hypothesis:

H2a: During equity incentive periods, the negative impact of executives' accounting competence on the relationship between goodwill impairment signal and goodwill impairment is stronger.

Furthermore, research has found that compared to companies recognizing goodwill impairment, avoidance of impairment resulting from mergers and acquisitions significantly increases the risk of stock price collapse. In addition, executives with equity incentives have their rewards tied to future stock prices. Therefore, management with higher long-term incentives and abilities are more motivated to enhance company performance and mitigate personal interest losses from stock price declines. Equity incentives can alleviate agency issues between owners and operators and enhance firm value [31]. Their study demonstrates that equity incentives serve as motivators. Therefore, I predict that, compared to periods without equity incentives, when in a period with equity incentives, executives with higher accounting abilities will not engage in opportunistic behavior when making the accounting choice of recognizing goodwill impairment, thus indicating a weaker negative impact of executives' accounting competence on the relationship between goodwill impairment signal and goodwill impairment.

Based on the comprehensive analysis, this study further proposes the following hypothesis:

H2b: During equity incentive periods, the negative impact of executives' accounting competence on the relationship between goodwill impairment signal and goodwill impairment is weaker.

## 3.3 Executives' accounting competence, key audit matters and goodwill impairment

A plethora of research has detected opportunistic behavior by executives in recognizing impairment of goodwill. In response to this issue, scholars have conducted studies from a regulatory constraint perspective. Initially, in terms of internal control mechanisms, Verriest and Gaeremynck examined the relationship between corporate governance mechanisms and goodwill impairment from the perspective of independent directors. Their research revealed a positive correlation between the level of corporate governance mechanisms and goodwill impairment, as supervisory mechanisms inhibit executives from utilizing goodwill impairment for earnings management behavior [10]. Corporate governance has a positive influence on firm performance and financial reporting quality [32–35]. Mansour, Amosh, Alodat, Khatib and Saleh explored that capital structure has a contingent role in the relationship between corporate governance quality and firm performance [36]. Saleh and Mansour found that multiple directorships lead to a scarcity of time that can adversely affect efficient management oversight [37]. With stricter internal constraints, executives have less discretion, and even executives with strong accounting capabilities will be discouraged from opportunistic behavior due to the higher implementation difficulty. As a result, I predict that when internal constraints are

stronger compared to weaker internal constraint mechanisms, the negative impact of executives' accounting competence on the goodwill impairment signal will be weaker.

Based on the comprehensive analysis, this study further presents the following hypothesis:

H3: When internal control mechanisms are stronger, the negative impact of executives' accounting competence on the relationship between goodwill impairment signal and goodwill impairment is weaker.

Furthermore, from the perspective of external supervision mechanisms, in countries with a more perfect judicial system, executives are more inclined to recognize goodwill impairment in a timely manner. Gietzmann and Wang found that when listed companies engage independent valuation experts, executives recognize goodwill impairment more promptly and for larger amounts [11]. Abhishek and Rajesh demonstrated that analyst coverage and audit quality affect the disclosure of goodwill impairment [12]. Within the external supervisory mechanism of accounting information, auditing is the most direct link. The subjectivity of goodwill recognition and impairment prompts auditors to pay special attention to the significant misstatement risk related to goodwill, leading to a significant investment of effort in designing and implementing audit procedures for goodwill [27]. Lin found that KAMs disclosure can improve audit quality [38]. Alshdaifat, Hamid, Saidin, and Aziz highlighted the importance of KAM, emphasizing its benefits in enhancing transparency [39]. According to our analysis, A-share listed companies disclosed 21 key audit matters related to goodwill impairment in 2016 (out of a total of 237 key audit matters disclosed that year), accounting for the highest proportion among all types of matters (8.86%), which increased to 14.45% in 2018. This indicates a high level of attention from auditors to goodwill impairment issues. When auditors list goodwill impairment as a key audit matter, even executives with strong accounting capabilities have a higher probability of having their opportunistic behavior exposed, thereby inhibiting such behavior. Therefore, I predict that compared to when goodwill impairment is not listed as a key audit matter, when auditors do list goodwill impairment as a key audit matter, the negative impact of executives' accounting competence on the goodwill impairment signal will be weaker.

Based on the comprehensive analysis, this study further presents the following hypothesis:

H4: When auditors designate goodwill impairment as a key audit matter, the negative impact of executives' accounting competence on the relationship between goodwill impairment signal and goodwill impairment is weaker.

## 4 Research design

### 4.1 Sample selection and data source

The study used A-share listed companies that have signed performance compensation commitment agreements from 2007 to 2022 as the initial sample. It excluded shell listing samples and retained only those where the listed company was the acquiring party. The goodwill data utilized in the study was sourced from the CSMAR database, and the performance compensation commitment data came from the WIND database. Some missing data was manually supplemented through publicly disclosed performance compensation commitment announcements and annual reports of the listed companies. After removing samples with missing data, the study obtained a total of 518 M&A events, comprising 1697 annual target data. To mitigate the influence of extreme values, the study subjected all continuous variables to a 1% winsorization process.

## 4.2 Variable definition

(1) **Explained variable.** Goodwill impairment. The article employs both dummy variables and continuous variables to measure goodwill impairment. *GWI_D* indicates whether the management recognizes goodwill impairment in year t; it is equal to 1 if goodwill impairment is recognized, and 0 otherwise. *GWI_LNA* is equal to the natural logarithm of (the amount of goodwill impairment loss recognized in year t + 1).

(2) **Explanatory variable.** Goodwill impairment signal. The degree of performance compensation commitment realization (*PROPMI*) was used as a goodwill impairment signal to measure goodwill impairment risk.

(3) **Moderating variable.** Executives' accounting competence. If the executives' professional background is related to accounting, *ACCCOMPET* is equal to 1; otherwise, it is equal to 0.

(4) **Grouping variable.** Equity Incentives. Referring to the practices of existing research, for the annual samples during the implementation of incentive equity incentives, *EI* is equal to 1, and for the annual samples of equity-incentive enterprises in other years and the annual samples of enterprises that have not implemented equity incentives, *EI* is equal to 0. Key Audit Matters. The auditor conducted a key audit matter audit on the impairment of goodwill, *KAM* is equal to 1; otherwise, it is equal to 0.

(5) **Control variable.** Referring to existing research, the control variables selected in this paper include the following variables: company performance, earnings management motivation, debt and compensation contracts, executive characteristics, and the degree of target asset performance commitment realization. Company performance was measured by return on assets (*ROA*), annual stock returns (*RET*), and revenue growth rate (*GROWTH*). Earnings management motivation was measured by smoothing motive (*SMOOTH*) and "big bath" motive (*BATH*). Regarding debt and compensation contracts, the debt contract was measured by the asset-liability ratio (*LEV*), and the equity incentive compensation contract was measured by the manager's shareholding ratio (*MSHARE*). For executive characteristics, the variables of CEO or general turnover (*CEOTURNOVER*) and executive tenure (*TENURE*) were used for measurement. Finally, the natural logarithm of total assets (*SIZE*), initial balance of goodwill (*GW*), operating income (*SALE*), property rights nature (*GOV*), and institutional investor shareholding (*INST*).

## 4.3 Model selection

With reference to the studies of Beatty and Weber, Ramanna and Watts [6, 7], in order to test research hypotheses H1a and H1b, this study uses Model 1 for testing. Research hypotheses H2a, H2b, H3, and H4 are tested in groups based on Model 1. When the dependent variable is the goodwill impairment dummy variable (*GWI_D*), the Logit model is used, and when the dependent variable is the goodwill impairment continuous variable (*GWI_LNA*), the Tobit model is used. Table 1 presents the definitions of the relevant variables.

$$GWI\_D_{it}/GWI\_LNA_{it} = \beta_0 + \beta_1 ACCCOMPET_{it} + \beta_2 PROPMI_{it} +$$
$$\beta_3 PROPMI * ACCCOMPET_{it} + \beta CONTROLS_{it} + YEAR_{it} + IND_{it} + \varepsilon_{it} \tag{1}$$

## 5 Empirical results

### 5.1 Descriptive statistics

Table 2 displays the descriptive statistics of the variables in the regression models. The mean value of the dependent variable, the goodwill impairment dummy variable (*GWI_D*), is 0.117,

**Table 1. Variable definitions.**

| Variable type | Variable name | Variable symbol | Variable definition |
|---|---|---|---|
| Explained variable | Goodwill impairment | GWI_D | A dichotomous variable equal to one if the firm recorded a goodwill impairment, zero otherwise. |
| | | GWI_LNA | Equal to the natural logarithm of (Goodwill impairment loss + 1). |
| Explanatory variable | Proportion of performance compensation commitment | PROPMI | Equal to [(promised performance—actual performance) divided by Total assets]*100. |
| Moderating variable | Executives' accounting competence | ACCCOMPET | A dichotomous variable equal to one if manager's professional background is related to accounting, zero otherwise. |
| Grouping variable | Equity incentive | EI | A dichotomous variable equal to one if companies implementing equity incentives during the incentive period, zero otherwise. |
| | Key audit matters | KAM | A dichotomous variable equal to one if the auditor conducted a key audit matter audit on the impairment of goodwill, zero otherwise. |
| Control variable | Equity concentration | HLST | A dichotomous variable equal to one if when the shareholding proportion of the largest shareholder of the company is greater than the median in the sample, zero otherwise. |
| | Annual stock return rate | RET | Equal to the annual return in year t estimated by monthly stock returns after adjustment by market returns. |
| | Operating income growth rate | GROWTH | Equal to (current operating income—previous operating income) divided by previous operating income |
| | Return on total assets | ROA | Equal to net profit divided by total assets at the end of the year |
| | "Big bath" motivation | BATH | A dichotomous variable equal to one if ROA < 0 and ΔROA is less than the median of all negative values, zero otherwise. |
| | Earnings smoothing motivation | SMOOTH | A dichotomous variable equal to one if ROA > 0 and ΔROA is greater than the median of all positive values, zero otherwise. |
| | Debt covenant pressure | LEV | Equal to total liabilities divided by total assets at the end of the period |
| | Manager shareholding | MSHARE | Equal to total manager's shareholding divided by total capital stock |
| | CEO turnover | CEOTURNOVER | A dichotomous variable equal to one if the CEO changed in year t, zero otherwise. |
| | Managerial tenure | TENURE | Equal to manager's tenure |
| | Company size | SIZE | Equal to the ln of total assets of year t |
| | Goodwill ratio | GW | Equal to amount of goodwill divided by total assets of year t |
| | Operating income | SALE | Equal to operating income divided by total assets f year t |
| | Nature of property | GOV | A dichotomous variable equal to one if the property rights are state-owned, zero otherwise. |
| | Proportion of institutional investors holding shares | INST | Equal to the number of shares held by institutional investors divided by the total share |

indicating that the proportion of samples with recognized goodwill impairment is 11.7%. The mean value of the continuous variable of goodwill impairment (*GWI_LNA*) is 2.078, with a minimum value of 0 and a maximum value of 20.359, indicating significant differences in the scale of recognized goodwill impairment among the sample companies. The mean value of the performance compensation commitment realization degree (*PROPMI*) is -0.188, indicating that the performance compensation commitment of most target assets has been achieved. The greater the value, the lower the degree of achievement of the performance compensation commitment. The mean value of the Executives' accounting competence *(ACCCOMPET)* is 0.220, indicating that approximately 22% of the executives in the sample have a professional background related to accounting. The mean value of equity incentives (*EI*) is 0.384, indicating that approximately 38.4% of the sample companies are in an equity incentive plan period. The mean value of key audit matters (*KAM*) is 0.319, indicating that approximately 31.9% of the sample companies have the auditor listing the goodwill impairment issue as a key audit matter, reflecting a high level of attention from auditors to the goodwill impairment issue.

**Table 2. Descriptive statistics of variables.**

| Variables | Obs | Mean | SD | Min | Median | Max |
|---|---|---|---|---|---|---|
| GWI_D | 1697 | 0.117 | 0.322 | 0.000 | 0.000 | 1.000 |
| GWI_LNA | 1697 | 2.078 | 5.747 | 0.000 | 0.000 | 20.359 |
| PROPMI | 1697 | -0.188 | 1.398 | -7.834 | -0.077 | 5.214 |
| ACCCOMPET | 1697 | 0.220 | 0.414 | 0.000 | 0.000 | 1.000 |
| EI | 1697 | 0.384 | 0.487 | 0.000 | 0.000 | 1.000 |
| KAM | 1697 | 0.319 | 0.466 | 0.000 | 0.000 | 1.000 |
| HLST | 1697 | 0.501 | 0.500 | 0.000 | 1.000 | 1.000 |
| BATH | 1697 | 0.075 | 0.263 | 0.000 | 0.000 | 1.000 |
| SMOOTH | 1697 | 0.224 | 0.417 | 0.000 | 0.000 | 1.000 |
| RET | 1697 | 0.209 | 0.782 | -0.639 | -0.083 | 3.208 |
| GROWTH | 1697 | 0.543 | 0.865 | -0.469 | 0.320 | 5.540 |
| ROA | 1697 | 0.027 | 0.084 | -0.513 | 0.039 | 0.129 |
| CEOTURNOVER | 1697 | 0.261 | 0.439 | 0.000 | 0.000 | 1.000 |
| TENURE | 1697 | 5.340 | 3.424 | 0.000 | 5.000 | 14.000 |
| LEV | 1697 | 0.384 | 0.190 | 0.065 | 0.359 | 0.903 |
| MSHARE | 1697 | 0.215 | 0.184 | 0.000 | 0.212 | 0.620 |
| INST | 1697 | 0.198 | 0.164 | 0.002 | 0.148 | 0.688 |
| SIZE | 1697 | 22.123 | 0.854 | 20.340 | 22.064 | 24.755 |
| GW | 1697 | 0.168 | 0.168 | 0.000 | 0.120 | 0.942 |
| SALE | 1697 | 0.467 | 0.293 | 0.099 | 0.392 | 1.821 |
| GOV | 1697 | 0.104 | 0.305 | 0.000 | 0.000 | 1.000 |

## 5.2 Multivariate results

Table 3 reports the impact of executives' accounting competence on relationship between goodwill impairment signal and goodwill impairment. As shown in Table 3, the coefficient of the goodwill impairment signal (*PROPMI*) is significantly positive. This indicates that the higher the degree of non-achievement of the performance compensation commitment, the higher the probability and scale of goodwill impairment, which is consistent with previous research findings [12]. My focus is on the interaction term between executives' accounting competence (*ACCCOMPET*) and the goodwill impairment signal (*PROPMI*), and its relationship with the probability and scale of goodwill impairment. The coefficient of the interaction term (*PROPMI*ACCCOMPET*) is -0.796 and -3.092, which are significant at the 10% and 5% levels, respectively. This suggests that executives' accounting competence has a negative effect on relationship between goodwill impairment signal and goodwill impairment. Research hypothesis H1b is confirmed.

## 5.3 Further analysis

In the earlier discussion, I elaborated that in recent years, due to significant goodwill impairment recognition by many listed companies leading to a sharp decline in performance, goodwill impairment has drawn attention from various parties. To avoid the negative impact of opportunistic behavior exposure, executives' opportunistic behavior may become more concealed. Therefore, I predict that the stronger the executives' accounting competence, the more discrepant his discretionary choices will be in different intervals of performance compensation commitment realization. Firstly, when the performance compensation commitment of the target assets is fully achieved, as the risk of goodwill impairment is low, the incentive for

**Table 3. The impact of executives' accounting competence on relationship between goodwill impairment signal and goodwill impairment.**

| Variables | (1) | (2) |
|---|---|---|
| | GWI_D | GWI_LNA |
| ACCCOMPET | 0.010 | 0.305 |
| | (0.032) | (0.127) |
| PROPMI | 1.805*** | 9.194*** |
| | (5.706) | (9.695) |
| PROPMI*ACCCOMPET | -0.796* | -3.092** |
| | (-1.715) | (-2.095) |
| BATH | 1.817*** | 16.119*** |
| | (4.104) | (4.577) |
| SMOOTH | -0.151 | -2.097 |
| | (-0.436) | (-0.840) |
| RET | -1.092*** | -7.670*** |
| | (-2.917) | (-2.954) |
| GROWTH | -0.410** | -2.343 |
| | (-1.974) | (-1.453) |
| ROA | -0.146 | 14.396 |
| | (-0.096) | (1.278) |
| CEOTURNOVER | -0.283 | -2.503 |
| | (-1.018) | (-1.134) |
| TENURE | 0.043 | 0.477* |
| | (1.050) | (1.853) |
| LEV | -1.084 | -10.980** |
| | (-1.393) | (-1.973) |
| MSHARE | -0.205 | -6.825 |
| | (-0.225) | (-1.015) |
| INST | -1.219 | -11.075 |
| | (-1.311) | (-1.621) |
| SIZE | 0.372** | 2.671** |
| | (2.165) | (2.074) |
| GW | 1.407 | 4.179 |
| | (1.544) | (0.693) |
| SALE | 0.188 | 1.295 |
| | (0.479) | (0.371) |
| GOV | 0.170 | 2.729 |
| | (0.417) | (0.823) |
| _cons | -9.038** | -250.167 |
| | (-2.356) | (-0.001) |
| Year | Yes | Yes |
| Ind | Yes | Yes |
| N | 1561 | 1697 |
| Pseudo. $R^2$ | 0.459 | 0.194 |

Note: * * *, * *, * are significant at 1%, 5% and 10% respectively, t values in parentheses.

opportunistic behavior from the manager is weaker. The stronger the executives' accounting competence, the lower the likelihood of implementing opportunistic behavior in this interval. Secondly, based on Article 54 of China's "Measures for the Administration of Major Asset Restructuring of Listed Companies" issued in 2008, when the performance compensation

commitment is not fulfilled by 80%, the profitability of the target assets is low, and the risk of goodwill impairment is high. If the manager engages in opportunistic behavior at this time, the probability of being discovered is higher. In this interval, the probability of implementing opportunistic behavior may also be lower with a stronger executives' accounting competence. However, when the performance commitment realization degree falls in the 80% to 100% interval, on one hand, due to the unfulfilled performance commitment, the motivation for opportunistic behavior from the manager is strong. On the other hand, as the performance commitment realization degree reaches 80%, the profitability of the target assets is still considerable, and the likelihood of discovering opportunistic behavior from the manager is low. Therefore, the probability of executives implementing opportunistic behavior in this interval may be higher. Based on the above analysis, I predict that compared to the intervals of performance compensation commitment realization below 80% and above 100%, the negative effect of executives' accounting competence will be stronger when the performance compensation commitment realization falls in the 80% to 100% interval.

Table 4 reports the results of the grouped regression analysis on the impact of executives' accounting competence on relationship between goodwill impairment signal and goodwill impairment. As expected, the results align with my predictions. In Table 4, when the dependent variable is the goodwill impairment dummy variable (*GWI_D*), only in the 80% to 100% interval, the coefficient of the interaction term between executives' accounting competence and the goodwill impairment signal (*PROPMI*ACCCOMPET*) is significantly negative, while it is not significant in the other two groups. When the dependent variable is the continuous variable of goodwill impairment (*GWI_LNA*), in the groups with performance compensation commitment realization below 80% and in the 80% to 100% interval, the coefficient of the interaction term between executives' accounting competence and the goodwill impairment signal (*PROPMI*ACCCOMPET*) is significantly negative. Furthermore, Table 4 reports the intergroup difference in coefficients for the groups with performance compensation commitment realization below 80%, 80% to 100%, and above 100%. It is evident that, whether in terms of impairment probability or magnitude, the negative effect of manager's accounting ability on the goodwill impairment signal is stronger in the 80% to 100% interval of

**Table 4. Grouped test of the impact of executives' accounting competence on relationship between goodwill impairment signal and goodwill impairment.**

| Variables | (1) | (2) | (3) | (4) | (5) | (6) |
|---|---|---|---|---|---|---|
| | GWI_D <80% | GWI_D 80%~100% | GWI_D >100% | GWI_LNA <80% | GWI_LNA 80%~100% | GWI_LNA >100% |
| ACCCOMPET | 1.114 | 0.081 | 0.283 | 4.180 | 0.460 | 0.140 |
| | (0.803) | (0.113) | (0.299) | (1.133) | (0.072) | (0.950) |
| PROPMI | 0.405 | 3.355** | 3.070 | 1.032 | 25.847*** | 0.097* |
| | (1.270) | (2.573) | (1.076) | (1.330) | (3.045) | (1.940) |
| PROPMI*ACCCOMPET | -0.725 | -6.979*** | -2.638 | -2.361* | -49.720** | -0.074 |
| | (-1.370) | (-3.064) | (-0.889) | (-1.791) | (-2.465) | (-0.675) |
| Controls | Yes | Yes | Yes | Yes | Yes | Yes |
| Year | Yes | Yes | Yes | Yes | Yes | Yes |
| Ind | Yes | Yes | Yes | Yes | Yes | Yes |
| N | 160 | 190 | 638 | 195 | 233 | 1269 |
| Inter-group Coefficient | 0.054* | | 0.073* | | 0.039** | 0.045** |
| Pseudo. $R^2$ | 0.417 | 0.327 | 0.436 | 0.124 | 0.158 | 0.038 |

Note: The inter-group coefficient difference test reports p-values, which are consistent with the significance levels in the above table. The same applies to subsequent content.

performance compensation commitment realization. The regression results in Table 4 indicate that the stronger the executives' accounting competence, the more concealed their opportunistic behavior becomes.

### 5.4 Heterogeneity analysis

**5.4.1 The impact of internal incentive mechanisms.** Table 5 reports the impact of internal incentive mechanisms measured by equity incentives. Due to the complexity of considering three interaction terms, I used a grouping form for heterogeneity analysis. As shown in the results of Table 5, in columns (1) and (3), the coefficient of the interaction term (*PROPMI*ACCCOMPET*) between executives' accounting competence and impairment signals of goodwill is not significant for companies that implemented equity incentives during the sample period, and the negative effect of executives' accounting competence is suppressed. In columns (2) and (4), for companies that did not implement equity incentives during the sample period, the coefficient of the interaction term (*PROPMI*ACCCOMPET*) between executives' accounting competence and impairment signals of goodwill is significantly negative. Furthermore, the inter-group coefficient difference test indicates a significant difference in the impact of executives' accounting competence on impairment signals of goodwill between the two groups of whether equity incentives were implemented. The results in Table 5 indicate that equity incentives have a motivating effect on executives, which can suppress executives' opportunistic behavior. When in the period of equity incentives, the negative effect of executives' accounting competence is weaker. Research hypothesis H2b has been validated.

**5.4.2 The impact of internal constraint mechanisms.** Table 6 reports the impact of internal constraint mechanisms measured by equity concentration. As shown in Table 6, significant negative coefficients for the interaction term (*PROPMI*ACCCOMPET*) between executives' accounting competence and impairment signals of goodwill are only observed in columns (2) and (4), the low equity concentration group. However, the inter-group coefficient difference test between the low and high equity concentration groups indicates no difference in the impact of executives' accounting competence on impairment signals of goodwill. To ensure the robustness of the results, I also used the Herfindahl index to measure internal constraint mechanisms, and the results were consistent. Research hypothesis H3 was not validated. The

**Table 5. The impact of internal incentive mechanisms measured by equity incentives.**

| Variables | (1) | (2) | (3) | (4) |
|---|---|---|---|---|
| | GWI_D (EI = 1) | GWI_D (EI = 0) | GWI_LNA (EI = 1) | GWI_LNA (EI = 0) |
| ACCCOMPET | -3.157*** | 0.427 | -14.736** | 3.448 |
| | (-3.571) | (1.229) | (-2.246) | (1.227) |
| PROPMI | 1.412*** | 2.172*** | 7.188*** | 10.103*** |
| | (3.493) | (5.529) | (4.880) | (8.035) |
| PROPMI*ACCCOMPET | -0.796 | -1.145** | 0.986 | -3.585** |
| | (-1.064) | (-2.335) | (0.185) | (-2.159) |
| _cons | -18.514* | -6.566 | -250.099 | -222.915 |
| | (-1.776) | (-1.231) | (-0.001) | (-0.002) |
| Controls | Yes | Yes | Yes | Yes |
| Year | Yes | Yes | Yes | Yes |
| Ind | Yes | Yes | Yes | Yes |
| N | 478 | 934 | 652 | 1045 |
| Inter-group Coefficient | 0.041** | | 0.003*** | |
| Pseudo. $R^2$ | 0.512 | 0.480 | 0.251 | 0.203 |

**Table 6. The impact of internal constraint mechanisms measured by equity concentration.**

| Variables | (1) | (2) | (3) | (4) |
|---|---|---|---|---|
| | GWI_D | GWI_D | GWI_LNA | GWI_LNA |
| | (HLST = 1) | (HLST = 0) | (HLST = 1) | (HLST = 0) |
| ACCCOMPET | -0.162 | 0.069 | 0.058 | 0.200 |
| | (-0.328) | (0.176) | (0.012) | (0.071) |
| PROPMI | 1.547*** | 2.285*** | 9.712*** | 8.652*** |
| | (2.952) | (5.708) | (5.632) | (7.742) |
| PROPMI*ACCCOMPET | -0.260 | -1.178* | -2.643 | -3.453* |
| | (-0.418) | (-1.944) | (-1.091) | (-1.699) |
| Controls | Yes | Yes | Yes | Yes |
| Year | Yes | Yes | Yes | Yes |
| Ind | Yes | Yes | Yes | Yes |
| N | 681 | 774 | 851 | 846 |
| Inter-group Coefficient | 0.738 | | 0.229 | |
| Pseudo. $R^2$ | 0.537 | 0.502 | 0.262 | 0.198 |

results in Table 6 indicate that internal constraint mechanisms have a weaker inhibitory effect on executives' opportunistic behavior regarding goodwill impairment. To address this issue, more consideration should be given to internal incentive mechanisms.

**5.4.3 The impact of external supervision mechanisms.** Table 7 reports the impact of external supervision mechanisms measured by whether auditors list goodwill impairment issues as key audit matters. As shown in the results of columns (1) and (3) in Table 7, when auditors list goodwill impairment issues as key audit matters, the coefficient of the interaction term (*PROPMI*ACCCOMPET*) between executives' accounting competence and goodwill impairment signals is not significant. However, in columns (2) and (4), the coefficient of the interaction term (*PROPMI*ACCCOMPET*) between executives' accounting competence and goodwill impairment signals is significantly negative. Furthermore, the inter-group coefficient difference test between the two groups of whether auditors list goodwill impairment issues as

**Table 7. The impact of external supervision mechanisms measured by key audit matters.**

| Variables | (1) | (2) | (3) | (4) |
|---|---|---|---|---|
| | GWI_D (KAM = 1) | GWI_D (KAM = 0) | GWI_LNA (KAM = 1) | GWI_LNA (KAM = 0) |
| ACCCOMPET | -0.272 | -0.321 | 0.903 | -2.063 |
| | (-0.464) | (-0.766) | (0.313) | (-0.464) |
| PROPMI | 2.062*** | 1.912*** | 6.935*** | 13.884*** |
| | (3.658) | (4.616) | (6.596) | (6.181) |
| PROPMI*ACCCOMPET | 0.992 | -1.513** | -0.908 | -9.209*** |
| | (0.959) | (-2.559) | (-0.551) | (-2.819) |
| _cons | -13.594* | -5.780 | -130.637 | -262.707 |
| | (-1.726) | (-0.970) | (-0.030) | (-0.001) |
| Controls | Yes | Yes | Yes | Yes |
| Year | Yes | Yes | Yes | Yes |
| Ind | Yes | Yes | Yes | Yes |
| N | 513 | 971 | 542 | 1155 |
| Inter-group Coefficient | 0.004*** | | 0.001*** | |
| Pseudo. $R^2$ | 0.527 | 0.385 | 0.177 | 0.194 |

key audit matters indicate a significant difference in the impact of executives' accounting competence on goodwill impairment signals. The results in Table 7 indicate that auditors' attention to goodwill impairment issues has a supervisory effect on executives, which can suppress executives' opportunistic behavior. When auditors list goodwill impairment issues as key audit matters, the negative effect of executives' accounting competence is weaker. Research hypothesis H4 has been validated.

## 5.5 Robustness test

**5.5.1 Endogenous test.** The research examines the potential for self-selection bias in the signing of performance-based compensation agreements by both parties in the acquisition and sale of target assets. This study controls for potential endogeneity issues using the Heckman two-stage model. In the first stage, a model is established with the signing of performance-based compensation agreements ($PMI$) as the dependent variable. The transaction payment method ($PAYMED$) is used as an instrumental variable in the Probit regression, and the inverse Mills ratio ($IMR$) is calculated. If the transaction payment method ($PAYMED$) is in the form of equity, its value is set as 1; otherwise, it is set as 0. The transaction payment method ($PAYMED$) is likely to affect the probability of signing performance-based compensation agreements, but its direct impact on subsequent goodwill impairment is relatively small, aligning with the instrumental variable selection criteria. In the second stage, the calculated inverse Mills ratio ($IMR$) is incorporated into the corresponding model for regression analysis.

$$PMI = \beta_0 + \beta_1 PAYMED_{it} + \beta CONTROLS_{it} + YEAR_{it} + IND_{it} + \varepsilon_{it} \qquad (2)$$

Table 8 presents the results of the Heckman two-stage regression. As shown in Table 8, the regression results of the first stage indicate that the coefficient of the transaction payment method ($PAYMED$) is significantly positive, indicating the presence of self-selection issues in whether both parties in the acquisition and sale of target assets sign performance-based compensation agreements. The second-stage regression results show that, after controlling for the inverse Mills ratio ($IMR$), the regression coefficient of the interaction term between executives' accounting competence and goodwill impairment signals ($PROPMI*ACCCOMPET$) remains significantly negative, indicating the robustness of the results.

**5.5.2 Other robustness tests.** Based on previous studies, the article also conducted the following robustness tests.

(1) The measurement method of the explained variable ($GWI\_A$) has been changed. Total assets are used as the adjustment standard, and the impairment amount of goodwill is divided by the initial total assets to measure the scale of goodwill impairment. The regression results are reported in the first column of Table 9, which are generally consistent with the existing conclusions.

(2) The measurement method of the explanatory variable ($MA/MA4$) has been changed. Management ability is used to replace executives' accounting competence. Firstly, following the approach of Demerjian et al. [26], management ability ($MA$) is measured using the two-stage method of data envelopment analysis (DEA) and Tobit regression analysis. Secondly, the regression residuals are sorted into four groups from small to large, and the corresponding managerial ability ($MA4$) is assigned values of 1, 2, 3, and 4 to measure it, with a higher value indicating stronger managerial ability. The regression results are reported in the second and third columns of Table 9, which are generally consistent with the existing conclusions. The

**Table 8. Heckman two-stage regression results.**

| Variables | The first stage | The second stage | |
|---|---|---|---|
| | (1) | (2) | (3) |
| | PMI | GWI_D | GWI_LNA |
| ACCCOMPET | | -0.069 | -0.979 |
| | | (-0.180) | (-0.333) |
| PROPMI | | 2.179*** | 10.097*** |
| | | (4.608) | (8.369) |
| PROPMI*ACCCOMPET | | -1.170* | -4.042** |
| | | (-1.819) | (-2.292) |
| PAYMED | 0.422** | | |
| | (2.410) | | |
| IMR | | -0.376 | -0.496 |
| | | (-0.333) | (-0.130) |
| Controls | Yes | Yes | Yes |
| Year | Yes | Yes | Yes |
| Ind | Yes | Yes | Yes |
| N | 686 | 1047 | 1172 |
| Pseudo. $R^2$ | 0.544 | 0.487 | 0.208 |

model for calculating management ability (MA) is as follows:

$$\max \theta t = \frac{Sales_t}{V_1 COGS_t + V_2 S\&M_t + V_3 PPE_{t-1} + V_4 Intang_{t-1} + V_5 R\&D_{t-1} + V_6 GW_{t-1}} \tag{3}$$

**Table 9. Robustness test—Replace variables.**

| Variables | (1) | (2) | (3) |
|---|---|---|---|
| | GWI_A | GWI_A | GWI_A |
| PROPMI | 2.878*** | 2.593*** | 3.380*** |
| | (10.499) | (9.928) | (6.006) |
| ACCCOMPET | 0.317 | | |
| | (0.441) | | |
| PROPMI*ACCCOMPET | -1.402*** | | |
| | (-3.375) | | |
| MA | | 5.302** | |
| | | (2.055) | |
| PROPMI*MA | | -2.464* | |
| | | (-1.909) | |
| MA4 | | | 0.724** |
| | | | (2.246) |
| PROPMI*MA4 | | | -0.330* |
| | | | (-1.666) |
| Controls | Yes | Yes | Yes |
| Year | Yes | Yes | Yes |
| Ind | Yes | Yes | Yes |
| N | 1172 | 1106 | 1106 |
| Pseudo. $R^2$ | 0.348 | 0.334 | 0.334 |

'Sales' refers to the company's operating income, representing its output. Also, COGS (cost of goods sold), S&M (selling and management expenses), PPE (property, plant, and equipment), Intang (intangible assets), R&D (research and development expenses), and GW (goodwill) collectively represent the company's input.

$$\theta t = \alpha_0 + \alpha_1 Size_t + \alpha_2 MS_t + \alpha_3 FCF_t + \alpha_4 Age_t$$
$$+\alpha_5 SOE_t + \alpha_6 \sum Year_t + \alpha_7 \sum Industry_t + \varepsilon_t \tag{4}$$

In this case, the company's characteristic variables include company size (*Size*), total operating revenue (*MS*), free cash flow (*FCF*), the logarithm of the number of years listed (*Age*), and property ownership (*SOE*). Additionally, the model incorporates industry and yearly fixed effects. The regression residuals $\varepsilon_t$ obtained from Model (5) represent management ability (*MA*).

(3) Due to the relatively limited sample of executives with accounting abilities, these samples may exhibit distinct characteristics. To control for these differences, in addition to the complete sample, I also utilized the Propensity Score Matching (PSM) sample. I matched samples of executives with and without accounting abilities and re-evaluated the impact of management accounting ability on goodwill impairment signals based on the paired samples. The regression results, as reported in Table 10, remain largely consistent with the existing conclusions.

## 6 Conclusions

Inconsistent with the general belief that highly competent executives provide better assurance for a company's earnings quality [24], enhance company value [18], secure higher acquisition premiums [21], improve stock performance, obtain higher credit ratings [27], and aid in tax planning [28], Albrecht, Mauldin and Newton found that when executives are motivated to disclose inaccurate information, those with strong accounting competence and extensive financial reporting experience may increase the likelihood of significant misreporting, revealing the dark side of executives' accounting competence in the presence of certain incentive factors [5]. Given that goodwill impairment reduces the net profit of listed companies and substantial goodwill impairment would severely impact the current performance of a listed company, often directly linked to executives' interests, executives are motivated to intervene in goodwill impairment for their own benefits [6–8]. Consequently, excellent executives'

**Table 10. Robustness test -PSM results.**

| Variables | (1) | (2) | (3) |
|---|---|---|---|
| | GWI_D | GWI_LNA | GWI_A |
| ACCCOMPET | 0.361 | 2.057 | 1.165 |
| | (0.632) | (0.630) | (1.302) |
| PROPMI | 2.990** | 13.176*** | 3.464*** |
| | (2.541) | (5.398) | (5.645) |
| PROPMI*ACCCOMPET | -2.104* | -8.824*** | -2.629*** |
| | (-1.785) | (-3.314) | (-3.888) |
| Controls | Yes | Yes | Yes |
| Year | Yes | Yes | Yes |
| Ind | Yes | Yes | Yes |
| N | 488 | 599 | 599 |
| Pseudo. $R^2$ | 0.539 | 0.252 | 0.401 |

accounting competence may exhibit negative effects in the context of goodwill impairment issues.

The study found that executives' accounting competence has a negative effect on relationship between goodwill impairment signal and goodwill impairment. It showed that the dark side of executives' accounting competence would appear in the presence of some incentive factors. This study enriches existing literature on the influencing factors of goodwill impairment, it provides a new perspective to test the opportunistic behavior of executives to accrue goodwill impairment. The existing literature of effect of goodwill impairment is mainly carried out from the perspectives of macroeconomic factors and discretion, and has achieved rich research results. But less attention is paid to the influence of executives' competence (especially accounting competence) on goodwill impairment. Similarly, it enriches the research literature on the characteristics of executives. Taking the performance compensation commitment as the breakthrough point, this paper examines the influence of executives' accounting competence on relationship between goodwill impairment signal and goodwill impairment and whether executives' accounting competence plays a positive or negative role, which expands the boundary of the literature on the characteristics of executives' competence.

Heterogeneity analysis showed that during stock incentive periods and when auditors perform key audit procedures for goodwill impairment, the negative impact of executives' accounting competence is dampened. The research found that internal incentive mechanisms and external supervision play a stronger role in mitigating executives' opportunistic behavior related to recognizing goodwill impairment, while the role of internal constraint mechanisms is weaker. It offers insights into mitigating executives' opportunistic behavior related to recognizing goodwill impairment, contributing to the formulation of accounting standards and assisting in capital market supervision.

Therefore, this study also provides significant practical implications for helping the regulatory agencies to strengthen the supervision of executives' behavior. Based on the impairment of goodwill in M&A, this paper points out that executives have the motivation, opportunity and competence to intervene in the impairment of goodwill for their own interests, and the accounting competence of executives have a negative effect, suggesting that regulators need to strengthen supervision over executives' behavior, which is crucial for upholding the normal functioning of the capital market.

## Author Contributions

**Writing – original draft:** Qiandan Deng.

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
