## [Decision Letter · Decision Letter 0]

20 Sep 2024

PONE-D-24-23860It's Dark Under the Lamp? The Influence of Executives' Accounting Competence on Goodwill Impairment SignalPLOS ONE

Dear Dr. DENG,

Thank you for submitting your manuscript to PLOS ONE. After careful consideration, we feel that it has merit but does not fully meet PLOS ONE’s publication criteria as it currently stands. Therefore, we invite you to submit a revised version of the manuscript that addresses the points raised during the review process.

We look forward to receiving your revised manuscript.

Kind regards,

Saleh F. A. Khatib, PhD

Guest Editor

PLOS ONE

“Sichuan Normal University Research Fund (Project No.: 22XW045)”

3. In the online submission form, you indicated that your data is available only on request from a third party. Please note that your Data Availability Statement is currently missing the name of the third party contact or institution / contact details for the third party, such as an email address or a link to where data requests can be made. Please update your statement with the missing information.

Additional Editor Comments:

The reviewers have raised a few concerns about the research quality, which I found to be valid upon reading the manuscript. These issues must be addressed before a final decision can be made;

Proofreading: The manuscript needs careful proofreading to address several language and formatting issues.

Literature Support: The majority of the discussion is not sufficiently supported by relevant references. Please enhance the literature review and update it with the most recent studies related to your research topic.

Introduction: The research rationale presented in the introduction needs improvement. It is essential to clearly articulate the research justification and align it with the specific context of your study.

Modeling: The models presented, specifically Model 1 and Model 2, could be merged into a comprehensive model, especially when considering the overall outcome variable (Y), to simplify the analysis and strengthen the narrative.

How did the authors address the endogeneity issue. refer to this study;An assessment of methods to deal with endogeneity in corporate governance and reporting research” Corporate Governance

Results: Your results section would benefit from being supported by prior research and theoretical frameworks. This will ensure that your findings are positioned within the existing body of knowledge.

Corrupt practice and sustainability reporting: Lifecycle perspective

Implications: The implications of your study, both theoretical and practical, should be thoroughly discussed. This will enhance the significance of your research contributions to both academic and practical domains.

Please make the necessary revisions and resubmit your manuscript. I look forward to seeing the improvements.

Reviewers' comments:

Reviewer's Responses to Questions

**Comments to the Author**

1. Is the manuscript technically sound, and do the data support the conclusions?

Reviewer #1: Partly

Reviewer #2: Partly

2. Has the statistical analysis been performed appropriately and rigorously? 

Reviewer #1: Yes

Reviewer #2: Yes

3. Have the authors made all data underlying the findings in their manuscript fully available?

Reviewer #1: Yes

Reviewer #2: Yes

4. Is the manuscript presented in an intelligible fashion and written in standard English?

Reviewer #1: Yes

Reviewer #2: Yes

5. Review Comments to the Author

Reviewer #1: Review of “It’s Dark Under the Lamp? The Influence of Executives' Accounting Competence on Goodwill Impairment Signal”

PONE-D-24-23860

Minor Revision

This paper aims to examines the influence of executives' accounting competence on the signal of goodwill impairment, considering the perspective of performance compensation commitment. The research employs an empirical research method and utilizes a sample of A-share listed companies in China that have signed performance compensation commitment agreements from 2007 to 2022 While it delivers novel theoretical conclusions, the study requires improvements and hence Minor issues should be addressed to improve the quality of the findings. kindly see my comments below:

Introduction

• In the introduction, the research motivation is unclear. Why now? Why China? Etc

• What is the research gap and how does the current study try to fill it? The authors should review prior studies to highlight the gaps they identify, and articulate how their study is distinct.

• You mentioned the research problem, but how to reach your answer is unclear.

Theoretical framework

• Please enhance your hypotheses by: (i) drawing on the theory; (ii) empirical literature; (iii) research setting/contextual insights; and (iv) then setting up your hypotheses. You will do this for each hypothesis. Currently, you have not developed your hypotheses in this way.

• The most recent studies should improve the theoretical issues and literature sections; some are listed here, please add to your list of citations:

Alshdaifat, S. M., Hamid, M. A. A., Saidin, S. F., & Ab Aziz, N. H. Insight of ISA 701: Key Audit Matter Disclosure in Extended Audit Report.‏14, 2, 278 – 287.

Alshdaifat, S.M., Abdul Hamid, M.A., Ab Aziz, N.H., Saidin, S.F. and Alhasnawi, M.Y. (2024), "Corporate governance effectiveness and firm performance in global crisis: evidence from GCC countries", Corporate Governance, Vol. ahead-of-print No. ahead-of-print. https://doi.org/10.1108/CG-12-2023-0518

Ab Aziz, N. H., Latiff, A. R. A., Alshdaifat, S. M., Osman, M. N. H., & Azmi, N. A. (2023). ESG Disclosure and Firm Performance: Evidence after the Revision of Malaysian Code of Corporate Governance. Int. J. Acad Res Bus Soc sci, 13, 12.‏

Mansour, M., Al Amosh, H., Alodat, A. Y., Khatib, S. F., & Saleh, M. W. (2022). The relationship between corporate governance quality and firm performance: The moderating role of capital structure. Sustainability, 14(17), 10525.‏

Saleh, M.W.A. & Mansour, M. (2024), "Is audit committee busyness associated with earnings management? The moderating role of foreign ownership", Accounting Research Journal, Vol. ahead-of-print No. ahead-of-print.https://doi.org/10.1108/ARJ-04-2023-0106

Mansour, M., Al Zobi, M., Al-Naimi, A., & Daoud, L., (2023). The connection between Capital structure and performance: Does firm size matter? Investment Management and Financial Innovations, 20(1), 195-206. https://doi:10.21511/imfi.20(1).2023.17

Research design

The research design is sufficiently robust.

Descriptive Statistics

The descriptive statistics is sufficiently robust.

Conclusions, Implications, and Recommendations

The conclusion section repeats the findings of the study, but it needs much more discussion of the implications of the findings for both academia and practice.

General comment

Please cite at least three papers related to the topic from the PLOS ONE.

Reviewer #2: 1. Hypothesis H2b proposes that due to the existence of equity incentives, executives with strong accounting capabilities will work harder to improve performance and will not engage in opportunistic behavior when making accounting choices to confirm goodwill impairment, thus indicating that management accounting capabilities have an impact on business performance. The negative impact of reputation impairment signals is weak. The logic of this assumption is not strong: first, there will be no opportunistic behavior when confirming goodwill impairment, that is, the goodwill impairment will be truthfully disclosed. However, if executives just want to improve performance and avoid losses from falling stock prices, disclosing the impairment of goodwill will also cause the stock price to fall and cause losses. Therefore, it is suggested that the supervisory role of equity incentives on executives can be supplemented to reduce speculative behavior.

2. Descriptive statistics say that the larger the PROPM value, the higher the degree of fulfillment of performance commitments. However, PROPM is defined as promised performance-actual performance. The smaller the value, the higher the degree of fulfillment.

3. Regression Table 3 shows the impact of the interaction term between accounting ability and goodwill impairment signal on goodwill impairment. The coefficient of accounting ability on goodwill impairment is not significant, and the coefficient of impairment signal and impairment is significantly positive. , the coefficient of the interaction term is negative. This regression uses a moderating effect model, but it is concluded that there is a relationship between accounting capabilities and goodwill impairment signals. Accounting capabilities and goodwill impairment signals are equivalent to two X's in this model. The moderating variable model cannot It is deduced that there is a relationship between the two X, and there is an error in the model and the empirical analysis in 5.2.

4. The conclusion should summarize the results of your article. You can use a small amount of text to introduce the conclusion of the article. However, there is a paragraph in the conclusion of this article that summarizes the literature and introduces the conclusion. It is recommended that it be streamlined.

6. PLOS authors have the option to publish the peer review history of their article (what does this mean?). If published, this will include your full peer review and any attached files.

Reviewer #1: No

Reviewer #2: No

---

## [Author Response · Author response to Decision Letter 0]

30 Oct 2024

Academic Editor

1、Proofreading: The manuscript needs careful proofreading to address several language and formatting issues. 

Thank you very much for your valuable advice.

I have carefully proofread the language and formatting issues of my manuscript to meet PLOS ONE's style requirements.

2、Literature Support: The majority of the discussion is not sufficiently supported by relevant references. Please enhance the literature review and update it with the most recent studies related to your research topic.

Thank you very much for your valuable advice. 

I have added the literature related to the research topic of my paper in the literature review, and the modified contents have been marked in the revised manuscript.

3、Introduction: The research rationale presented in the introduction needs improvement. It is essential to clearly articulate the research justification and align it with the specific context of your study. 

Thank you very much for your valuable advice. 

In the introduction, I added an explanation of the research rationale of my manuscript, and made it consistent with the specific context of my study. 

The modified contents have been marked in the revised manuscript.

4、Modeling: The models presented, specifically Model 1 and Model 2, could be merged into a comprehensive model, especially when considering the overall outcome variable (Y), to simplify the analysis and strengthen the narrative.

How did the authors address the endogeneity issue. refer to this study; An assessment of methods to deal with endogeneity in corporate governance and reporting research” Corporate Governance

Thank you very much for your valuable advice. Because the dependent variables of model 1 and model 2 are dummy variables and continuous variables of goodwill impairment, logit and tobit models are used respectively in the test, so two models are used in this paper. However, considering the simplification problem you mentioned, I finally decided to merge the two models into a comprehensive model, and explained the different test methods in the empirical part later.

In part 5.5.1, Heckman two-stage model is used to address the possible endogenous issue in my paper. After testing, the conclusion is robust. 

The modified contents have been marked in the revised manuscript.

5、Results: Your results section would benefit from being supported by prior research and theoretical frameworks. This will ensure that your findings are positioned within the existing body of knowledge. Corrupt practice and sustainability reporting: Lifecycle perspective.

Thank you very much for your valuable advice. My research is based on the previous research to discuss the issue of goodwill impairment, but my paper also has some supplements to the existing research, which is expounded in the research contribution of my paper. 

For example, Albrecht et al. (2018) found that when executives are motivated to disclose inaccurate information, those with strong accounting competence and extensive financial reporting experience may increase the likelihood of significant misreporting. Although we have all studied the accounting competence of executives, we are aiming at different problems.

The research contribution of my manuscript is mainly reflected in the following aspects. First, it provides a new perspective to test the opportunistic behavior of executives to accrue goodwill impairment. Secondly, it enriches the research literature on the characteristics of executives. Third, it will help the regulatory agencies to strengthen the supervision of executives' behavior.

6、 Implications: The implications of your study, both theoretical and practical, should be thoroughly discussed. This will enhance the significance of your research contributions to both academic and practical domains.

Thank you very much for your valuable advice. In the part of research contribution, I strengthened the theoretical contribution and practical contribution of my research. 

The modified contents have been marked in the revised manuscript.

Reviewer #1: 

1、Introduction

•In the introduction, the research motivation is unclear. Why now? Why China? Etc

Thank you very much for your valuable advice. 

This study is based on the data of Chinese enterprises. Although it will be influenced by some unique Chinese institutional backgrounds, as an outstanding representative of emerging countries, China's corporate governance problems under the Chinese scenario are also likely to exist in other emerging markets. What's more, the agency problem is a common problem faced by all markets. The specific research object of this article is goodwill impairment. With the rapid economic development, the upsurge of corporate mergers and acquisitions is playing out in many markets, and the large amount of goodwill and goodwill impairment risk are unavoidable problems in various markets. Therefore, some research conclusions obtained in this study may also be used for reference in other emerging markets.

In the introduction, the modified contents have been marked in the revised manuscript.

• What is the research gap and how does the current study try to fill it? The authors should review prior studies to highlight the gaps they identify, and articulate how their study is distinct.

Thank you very much for your valuable advice. In the research contribution section, I have answered these questions you raised. 

The modified contents have been marked in the revised manuscript.

• You mentioned the research problem, but how to reach your answer is unclear.

Thank you very much for your valuable advice. 

I revised and expounded how to solve the research problems in this paper, which made you more clear. 

“By examining the impact of executives' accounting competence on relationship between goodwill impairment signal and goodwill impairment, I can assess whether executives' strong accounting competence raises the likelihood of opportunistic behavior in the context of goodwill impairment. When the corporate governance mechanism is more effective, executives are less likely to act opportunistically and their discretion is more limited. Therefore, considering the corporate governance mechanism, we examine strategies to mitigate executives' opportunistic behavior in recognizing goodwill impairment.

The modified contents have been marked in the revised manuscript.

2、Theoretical framework

• Please enhance your hypotheses by: (i) drawing on the theory; (ii) empirical literature; (iii) research setting/contextual insights; and (iv) then setting up your hypotheses. You will do this for each hypothesis. Currently, you have not developed your hypotheses in this way.

Thank you very much for your valuable advice. 

I have developed my research hypothesis in the way you suggested. 

The modified contents have been marked in the revised manuscript.

• The most recent studies should improve the theoretical issues and literature sections; some are listed here, please add to your list of citations:

Alshdaifat, S. M., Hamid, M. A. A., Saidin, S. F., & Ab Aziz, N. H. Insight of ISA 701: Key Audit Matter Disclosure in Extended Audit Report.‏14, 2, 278 – 287.

Alshdaifat, S.M., Abdul Hamid, M.A., Ab Aziz, N.H., Saidin, S.F. and Alhasnawi, M.Y. (2024), "Corporate governance effectiveness and firm performance in global crisis: evidence from GCC countries", Corporate Governance, Vol. ahead-of-print No. ahead-of-print. https://doi.org/10.1108/CG-12-2023-0518

Ab Aziz, N. H., Latiff, A. R. A., Alshdaifat, S. M., Osman, M. N. H., & Azmi, N. A. (2023). ESG Disclosure and Firm Performance: Evidence after the Revision of Malaysian Code of Corporate Governance. Int. J. Acad Res Bus Soc sci, 13, 12.‏

Mansour, M., Al Amosh, H., Alodat, A. Y., Khatib, S. F., & Saleh, M. W. (2022). The relationship between corporate governance quality and firm performance: The moderating role of capital structure. Sustainability, 14(17), 10525.‏

Saleh, M.W.A. & Mansour, M. (2024), "Is audit committee busyness associated with earnings management? The moderating role of foreign ownership", Accounting Research Journal, Vol. ahead-of-print No. ahead-of-print.https://doi.org/10.1108/ARJ-04-2023-0106

Saleh, M.W.A. and Mansour, M. (2024), "Is audit committee busyness associated with earnings management? The moderating role of foreign ownership", Accounting Research Journal, Vol. 37 No. 1, pp. 80-97. https://doi.org/10.1108/ARJ-04-2023-0106

Mansour, M., Al Zobi, M., Al-Naimi, A., & Daoud, L., (2023). The connection between Capital structure and performance: Does firm size matter? Investment Management and Financial Innovations, 20(1), 195-206. https://doi:10.21511/imfi.20(1).2023.17

Marwan Mansour, Mo’taz Kamel Al Zobi, Ahmad Al-Naimi and Luay Daoud

(2023). The connection between Capital structure and performance: Does firm

size matter?. Investment Management and Financial Innovations, 20(1), 195-206.

doi:10.21511/imfi.20(1).2023.17

Thank you very much for your valuable advice. 

I have read these literatures and added them to the references. 

The modified contents have been marked in the revised manuscript.

3、Research design

The research design is sufficiently robust.

Thank you very much for your valuable advice.

4、Descriptive Statistics

The descriptive statistics is sufficiently robust.

Thank you very much for your valuable advice.

5、Conclusions, Implications, and Recommendations

The conclusion section repeats the findings of the study, but it needs much more discussion of the implications of the findings for both academia and practice.

Thank you very much for your valuable advice. 

In the conclusion section, I have simplified the description of the research findings of the paper, and strengthened the discussion of the academic and theoretical research significance of the research findings of the paper. 

The modified contents have been marked in the revised manuscript.

6、General comment

Please cite at least three papers related to the topic from the PLOS ONE.

Thank you very much for your valuable advice. 

I have read the following three literatures and added them to the references. 

The modified contents have been marked in the revised manuscript.

Lin J (2023) Does the disclosure of key audit matters improve the audit quality for sustainable development: Empirical evidence from China. PLoS ONE 18(5): e0285340. https://doi.org/10.1371/journal.pone.0285340

Sun H (2023) Corporate governance and reporting quality of accounts in China-listed firms. A moderating role of ownership pattern. PLoS ONE 18(11): e0295253. https://doi.org/10.1371/journal.pone.0295253

Arif HM, Mustapha MZ, Abdul Jalil A (2023) Do powerful CEOs matter for earnings quality? Evidence from Bangladesh. PLoS ONE 18(1): e0276935. https://doi.org/10.1371/journal.pone.0276935

Reviewer #2: 

1. Hypothesis H2b proposes that due to the existence of equity incentives, executives with strong accounting capabilities will work harder to improve performance and will not engage in opportunistic behavior when making accounting choices to confirm goodwill impairment, thus indicating that management accounting capabilities have an impact on business performance. The negative impact of reputation impairment signals is weak. The logic of this assumption is not strong: first, there will be no opportunistic behavior when confirming goodwill impairment, that is, the goodwill impairment will be truthfully disclosed. However, if executives just want to improve performance and avoid losses from falling stock prices, disclosing the impairment of goodwill will also cause the stock price to fall and cause losses. Therefore, it is suggested that the supervisory role of equity incentives on executives can be supplemented to reduce speculative behavior.

Thank you very much for your valuable advice.

The Research has found that compared to companies recognizing goodwill impairment, avoidance of impairment resulting from mergers and acquisitions significantly increases the risk of stock price collapse. In addition, executives with equity incentives have their rewards tied to future stock prices. Therefore, management with higher long-term incentives and abilities are more motivated to enhance company performance and mitigate personal interest losses from stock price declines. Equity incentives can alleviate agency issues between owners and operators and enhance firm value. Their study demonstrates that equity incentives serve as motivators.

In addition, my research also found that internal incentive mechanisms and external supervision play a stronger role in mitigating executives' opportunistic behavior related to recognizing goodwill impairment, while the role of internal constraint mechanisms is weaker.

Above all, I finally decided to study the impact of equity incentives from the perspective of incentives.

2. Descriptive statistics say that the larger the PROPM value, the higher the degree of fulfillment of performance commitments. However, PROPM is defined as promised performance-actual performance. The smaller the value, the higher the degree of fulfillment.

Thank you very much for your valuable advice.

In descriptive statistics, I described the value of performance compensation commitment from the perspective of treating it as a negative number. After checking, I found that it was wrong.

According to the definition, the greater the value, the lower the degree of achievement of the performance compensation commitment. Thank you very much for your suggestion. I have made some changes in the paper.

The modified contents have been marked in the revised manuscript.

3. Regression Table 3 shows the impact of the interaction term between accounting ability and goodwill impairment signal on goodwill impairment. The coefficient of accounting ability on goodwill impairment is not significant, and the coefficient of impairment signal and impairment is significantly positive. , the coefficient of the interaction term is negative. This regression uses a moderating effect model, but it is concluded that there is a relationship between accounting capabilities and goodwill impairment signals. Accounting capabilities and goodwill impairment signals are equivalent to two X's in this model. The moderating variable model cannot It is deduced that there is a relationship between the two X, and there is an error in the model and the empirical analysis in 5.2.

Thank you very much for your valuable advice.

This study is not intended to infer the relationship between two X's. I'm very sorry, there was something wrong with my previous expression, which caused you ambiguity.

According to my past research, the realization degree of performance compensation commitment can be used as a signal of whether goodwill needs to be depreciated. This study examines the influence of executives' accounting competence on relationship between goodwill impairment signal and goodwill impairment, considering the perspective of performance compensation commitment. The executives' accounting competence plays a moderating role. I have re-described my research problem in the article.

The modified contents have been marked in the revised manuscript.

4. The conclusion should summarize the results of your article. You can use a small amount of text to introduce the conclusion of the article. However, there is a paragraph in the conclusion of this article that summarizes the literature and introduces the conclusion. It is recommended that it be streamlined.

Thank you very much for your valuable advice.

In the conclusion section, I have simplified the description of the research findings of the paper, and strengthened the discussion of the academic and theoretical research significance of the research findings of the paper. 

The modified contents have been marked in the revised manuscript.

---

## [Decision Letter · Decision Letter 1]

15 Dec 2024

It's dark under the lamp? The moderating role of executives' accounting competence on relationship between goodwill impairment signal and goodwill impairment

PONE-D-24-23860R1

Dear Author,

We’re pleased to inform you that your manuscript has been judged scientifically suitable for publication and will be formally accepted for publication once it meets all outstanding technical requirements.

Kind regards,

Amira M. Idrees, Professor

Academic Editor

PLOS ONE

Additional Editor Comments (optional):

Reviewers' comments:

Reviewer's Responses to Questions

**Comments to the Author**

1. If the authors have adequately addressed your comments raised in a previous round of review and you feel that this manuscript is now acceptable for publication, you may indicate that here to bypass the “Comments to the Author” section, enter your conflict of interest statement in the “Confidential to Editor” section, and submit your "Accept" recommendation.

Reviewer #1: All comments have been addressed

Reviewer #2: All comments have been addressed

2. Is the manuscript technically sound, and do the data support the conclusions?

Reviewer #1: Yes

Reviewer #2: Yes

3. Has the statistical analysis been performed appropriately and rigorously? 

Reviewer #1: Yes

Reviewer #2: Yes

4. Have the authors made all data underlying the findings in their manuscript fully available?

Reviewer #1: Yes

Reviewer #2: Yes

5. Is the manuscript presented in an intelligible fashion and written in standard English?

Reviewer #1: Yes

Reviewer #2: Yes

6. Review Comments to the Author

Reviewer #1: (No Response)

Reviewer #2: After a long period of review and argumentation, it was found that the author has made careful revisions to address the issues raised last time, and has reached an average level above the published papers in the journal. Based on this, if there are no other special circumstances, the editorial department agrees to accept this manuscript.

7. PLOS authors have the option to publish the peer review history of their article (what does this mean?). If published, this will include your full peer review and any attached files.

Reviewer #1: No

Reviewer #2: No

---

## [Editor Report · Acceptance letter]

20 Dec 2024

PONE-D-24-23860R1 

PLOS ONE

Dear Dr. DENG, 

I'm pleased to inform you that your manuscript has been deemed suitable for publication in PLOS ONE. Congratulations! Your manuscript is now being handed over to our production team.

Kind regards, 

on behalf of

Prof. Amira M. Idrees 

Academic Editor

PLOS ONE